# Identification of Reference Genes for RT-qPCR Analysis in *Gleditsia microphylla* under Abiotic Stress and Hormone Treatment

**DOI:** 10.3390/genes13071227

**Published:** 2022-07-10

**Authors:** Jiaqi Yang, Fengying Han, Li Yang, Jin Wang, Feng Jin, An Luo, Fuyong Zhao

**Affiliations:** College of Life Sciences, Yangtze University, Jingzhou 434025, China; YJQ15002998506@163.com (J.Y.); hanfengying176@163.com (F.H.); yangli202203@126.com (L.Y.); wangjinformal@163.com (J.W.); jinjinfeng2021@126.com (F.J.); anluo@whu.edu.cn (A.L.)

**Keywords:** reference genes, *Gleditsia microphylla*, real-time quantitative PCR (RT-qPCR), abiotic stresses, hormone treatments

## Abstract

*Gleditsia microphylla* is an important galactomannan gums source plant with characteristics of drought resistance, barren tolerance, and good adaptability. However, the underlying molecular mechanisms of the biological process are not yet fully understood. Real-time quantitative PCR (RT-qPCR) is an accurate and convenient method to quantify the gene expression level and transcription abundance of suitable reference genes. This study aimed to screen the best internal reference genes in *G. microphylla* under abiotic stresses, hormone treatments, and different tissues. Based on the transcriptome data, twelve candidate reference genes were selected, and ultimately, nine of them were further evaluated by the geNorm, NormFinder, BestKeeper, and RefFinder algorithms. These results show that *TATA-binding protein 1* (*TBP1*)and *Eukaryotic translation initiation factor 4A1* (*EIF4A1*)were the two most stable reference genes, and *glyceraldehyde-3-phosphate dehydrogenase A subunit, chloroplastic* (*GAPA*)and *glyceraldehyde-3-phosphate dehydrogenase B subunit*, *chloroplastic* (*GAPB*)were the two most unstable reference genes across all samples under the given experimental conditions. Meanwhile, the most stable reference genes varied among the different groups and tissues. Therefore, this study suggests that it is better to use a specific reference gene for a particular case rather than using a common reference gene.

## 1. Introduction

Real-time quantitative PCR (RT-qPCR) is a technique that can precisely quantify nucleic acid molecules by monitoring fluorescence signals during the entire PCR process. It has been widely utilized in gene expression and transcript abundance analysis due to its high sensitivity, good repeatability, and strong specificity [1,2,3,4,5]. Although RT-qPCR is a powerful tool for understanding gene roles in metabolic pathways, signaling pathways, and complex regulatory networks in organisms, the normalization accuracy depends on suitable and stable reference genes as internal standards [6,7,8,9]. For an ideal reference gene used for normalization in real-time PCR analysis, its expression should remain constant between the cells of different tissues and under various experimental conditions. Commonly, the housekeeping genes involved in fundamental cellular processes such as *18S rRNA*, *ACT* (*actin-related protein*), *TUB* (*β-1 tubulin*), and *GAPDH* (*glyceraldehyde-3-phosphate dehydrogenase*) are used as reference genes in RT-qPCR for plants [2,3,4,5,6,7,8,9]. However, no universal reference genes with stable expression profiles in different tissues and organs, developmental stages, and experimental conditions have been discovered [10,11,12,13,14]. Therefore, it is vital to identify suitable internal reference genes to study the expression levels of target genes in different sample types, and under various experimental conditions. Several statistical algorithms such as geNorm [15,16], NormFinder [17], BestKeeper [18], and RefFinder [19] have emerged for screening stable internal reference genes for RT-qPCR normalization. Many novel reference genes have been identified and validated by employing these programs in different plants, such as *Rubia yunnanensis* [20], *Schima superba* [21], and *Piper* species [22].

*G. microphylla*, a shrub or tree species of the *Fabaceae* family, *Gleditsia* genus, is widely distributed in areas along the Taihang Mountains in northern China. It is a dioecious plant with a relatively long life span and commonly sets seeds in the third or fourth year after plantation [23,24]. The endosperm accounts for about 41.0% of the seed dry weight and is rich in galactomannan (>63.0%), the water solution of which can be used as a thickener, stabilizer, and adhesive in the oil drilling, food, medicine, printing, and dyeing industries. The embryo contains plentiful proteins (about 40.8%) and can be used as a nutrition ingredient for animal feed [25,26]. In addition, *G. microphylla* has many advantages, including drought resistance, cold resistance, barren tolerance, and a developed deep root system, which allow it to grow well under various stress conditions [23,25]. Owing to its economic and ecological benefits, exploring the biological characteristics of the plant at the molecular level is a worthy research field. 

However, as yet there are few reference genes that have been reported in the *Gleditsia* species. Therefore, twelve common housekeeping genes were selected as candidate reference genes from the transcriptome data and evaluated by the four popular software programs in this study, aiming to identify appropriate reference genes with stable expression for RT-qPCR analysis in different tissues (root, stem, leaf, flower) and under various experimental conditions (cold, heat, drought, salinity, and heavy-metal stresses; methyl jasmonate (MeJA), abscisic acid (ABA), and salicylic acid (SA) stimuli). To confirm the reliability of the selected reference genes, the best-ranked reference genes were further validated by normalizing the expression of *ACO1* (ethylene-forming enzyme) and *CSD2* (superoxide dismutase (Cu-Zn)), two stress-responding genes.

## 2. Materials and Methods

### 2.1. Plant Materials and Treatment

*G. microphylla* seeds were collected from Cixian County (N: 36°24′13.03″, E: 113°59′23.24″), Hebei Province, China. Healthy seeds were selected and germinated as previously described [27]. Then, they were sown into pots (3 seeds per pot) filled with a soil mixture and grown in a greenhouse (25 ± 1 °C, 16 h (L)/8 h (D) photoperiod, 3000-lux light intensity, 60–75% relative humidity). One and a half months later, the seedlings with uniform growth were rinsed well with tap water and then sequentially transferred to test tubes with distilled water, 1/2 strength, and Hoagland nutrient solution for two days each in the same greenhouse. For hormone stimulus treatments, the seedlings were spritzed with 100 μM methyl jasmonate (MeJA), 100 μM abscisic acid (ABA), or 200 μM salicylic acid (SA), while the control seedlings received only water. For heavy-metal and salinity stresses, the seedlings were treated with Hoagland solution supplemented with 200 μM copper sulfate (CuSO_4_) or 100 μM sodium chloride (NaCl). Samples of the top third and fourth leaves, stem, and root were collected 12 h after hormone, heavy-metal, and salt treatments. For cold and heat shock treatments, the plants were grown in a growth chamber at 4 °C or 42 °C for 24 h. For simulated drought treatment, the seedlings were subjected to Hoagland nutrient solution with 10% PEG6000 for 7 d. The mock-treated seedlings with the same time interval served as the control. Male and female flower samples were collected from adult plants in Jinzhou, Hubei Province. All the samples were frozen in liquid nitrogen and preserved at −80 °C.

### 2.2. Extraction of Total RNA and cDNA Synthesis

Total RNA was isolated from the frozen tissues using the *EasyPure*^®^ Plant RNA Kit (TransGen Biotech, Beijing, China) and quantified by a NanoDrop 2000c spectrophotometer (Thermo Scientific, Waltham, MA, USA). Only RNA samples with A_260_/A_280_ of 1.9–2.1 were used for cDNA synthesis. RNA integrity was analyzed by 1% agarose gel electrophoresis. The first-strand cDNA was synthesized with 1.0 μg total RNA in a 20 μL reaction system according to the *TransScript*^®^ All-in-One First-Strand cDNA Synthesis SuperMix for qPCR (One-Step gDNA Removal) Kit (TransGen Biotech, Beijing, China). All the cDNA samples were stored at −20 °C.

### 2.3. Screening of Candidate Reference Gene and Primer Design

Based on the transcriptome data of *G. microphylla* buds and flowers (http://www.ncbi.nlm.nih.gov/bioproject/848854 (14 June 2022)), a total of 12 internal housekeeping and 3 stress-responding genes cloned by our team with rapid amplification of cDNA ends (RACE) technique were selected in the present study (Appendix A). The homologs are commonly used as reference genes in other plant species and are extensively expressed in different tissues [2,3,4,5,6,7,8,9]. These genes are as follows: *actin 1* (*ACT1*, MZ210072), *actin 7* (*ACT7*, MZ210071), *tubulin β-1 chain 1* (*TUB1*, MZ210076), *GAPA* (MZ210073), *GAPB* (MZ210073), *TBP1* (MZ210062), *nuclear cap-binding protein 20* (*CBP20,* MZ210063, MZ210064), *TIP41-like protein* (*TIP41*, MZ210065, MZ210066), *translation elongation factor 1-α* (*EF-1α*, MZ210067, MZ210068), *EIF4A1* (MZ210069), *eukaryotic translation initiation factor 2* (*eIF2*, OL456245), and *polypyrimidine tract-binding protein 1* (*PTB1*, MZ210070). In addition, homologs of *1-aminocyclopropane-1-carboxylate oxidase 1* (*ACO1*, OL456246), *PYR1-like 1* (*PYL1*, OL456247), and *Copper/zinc superoxide dismutase 2* (*CSD2*, OL456248, OL456249) were utilized to validate the selected reference genes in further gene expression analysis. Specific primer pairs for all genes were designed with the online program Primer-BLAST (https://www.ncbi.nlm.nih.gov/tools/primer-blast/ (accessed on 10 June 2021)) and synthesized by Qingke Biotech (Wuhan, China). For analyzing the specificity of the designed primer pairs, the PCR products of each gene were checked by 2% agarose gel electrophoresis. In addition, the standard curves, melt curves, and PCR amplification efficiencies were determined before the RT-qPCR evaluation. The mixed cDNA from all samples was used as a template in primer evaluation. Standard curves of each primer pair were established using a 10-fold dilution series (10^0^, 10^−1^, 10^−2^, 10^−3^, and 10^−4^) of template cDNA. Melting curve analysis was followed by amplification in RT-qPCR. The amplification efficiency (*E*) of each primer pair and the correlation coefficient (*R*^2^) were generated automatically by CFX Manager. 

### 2.4. Real-Time PCR

A CFX96 Touch Real-Time PCR Detection System (Bio-Rad Laboratory, Inc., Hercules, CA, USA.) was used, and the program (3 min at 95 °C followed by 40 cycles at 94 °C for 15 s, and at 60.5 °C for 60 s) employed for RT-qPCR used a reaction mixture volume of 20 μL in an optical 96-well plate. Then, 10.0 μL of AceQ^®^ SYBR Green Master Mix (Vazyme Biotech, Nanjing, China), 0.3 μL of each final primer (150 nM), 2.0 μL of final cDNA (20 ng), and 7.4 μL RNase-free water were added to the reaction mixture. A control was also included in each plate with 2.0 μL of RNase-free water as a template. Two or three technical replicates were included for each biological replicate contained in each plate. The threshold cycle (Ct) values were calculated automatically by CFX Manager according to the overall expression levels of each gene analyzed. 

### 2.5. Statistical Analysis of Gene Expression and Comparison of Normalization Methods

The Ct values obtained from RT-qPCR for each candidate reference gene were inputted into software or an online website and analyzed according to the corresponding manuals of geNorm [15,16], NormFinder [17], BestKeeper [18], and RefFinder [19]. Briefly, when analyzed by geNorm and NormFinder, raw Ct values of each candidate must be converted to relative quantitative values (2^−ΔCt^), and an expression stability measurement (M) value and pairwise variation (V) value must be calculated by geNorm. All candidates were ranked by the M values, where the smaller the M value, the better the stability. The V value was used to determine the optimal numbers of reference genes, and it is generally considered that when the value of Vn/n + 1 is more than 0.15, the (n + 1)^th^ reference gene is in need. Compared with geNorm, NormFinder was used to compare the expression differences in candidate reference genes based on the calculated stability value. Similar to geNorm, the smaller the M value, the better the stability of the candidate; however, the NormFinder program can only select the most suitable gene as the internal reference gene. BestKeeper directly utilized the raw Ct value for stability analysis by calculating the coefficient of variance (CV) and the standard deviation (SD); the criteria for gene stability were smaller CV and SD values. Finally, the web-based tool RefFinder (http://blooge.cn/RefFinder/) (accessed on 1 May 2022) integrated all three methods mentioned above and raw Ct values to calculate the geometric mean for each reference gene and the comprehensive ranking index of stability. A lower index value indicates a higher stability of the candidate. 

### 2.6. Validation of Reference Genes by Expression Analysis of Stress-Responding Genes

Under stress conditions, ethylene, ABA, and reactive oxygen species (ROS) are essential signal transduction molecules to activate the defense system in plants, and ACO (ethylene-forming enzyme) [28], PYLs (abscisic acid receptors) [29,30], and CSD2 (superoxide radicals detoxifying enzyme) [31,32] play key roles during the process. To validate the selected reference genes in this study, the expression levels of these three homologs were analyzed using the most and least stable reference genes under stress conditions and calculated using the 2^−ΔΔCt^ method [33]. Three biological replicates were included for each treatment, and three technical replicates were included for each biological sample.

## 3. Results

### 3.1. Specificity of Primers for Candidate Reference Genes and Target Genes 

To detect the specificity of the designed primers for the twelve candidate reference genes and three target genes, analyses of the gel electrophoresis of the PCR products and melting curves were performed. The gel electrophoresis showed a single band with the expected size of each primer pair (Figure 1), and the melting curves of each primer pair exhibited a single peak (Appendix A), indicating the specificity of these primer pairs of candidate genes and target genes. The standard curves indicated that the RT-qPCR amplification efficiency of the candidate reference genes ranged from 86.90% (*TIP41*) to 118.60% (*CBP20*), and the correlation coefficients (*R*^2^) varied from 0.982 (*CBP20*) to 1.000 (*ACT7*) (Appendix A, Appendix A). Thus, all the primer pairs were specific for their respective genes and could be used in RT-qPCR analysis except *TIP41* and *ACT1* because of their lower amplification efficiencies. 

### 3.2. Expression Profiling of Candidate Reference Genes

Analyses of the expression levels of the remaining ten candidates were performed in all samples using RT-qPCR. The statistical results show that *PTB1* had the highest average Ct value of 32.08 among all the candidates, and the maximum values were over 35.00 in some root samples, which implied it was expressed at a low level and unsuitable to act as a normalization gene. Therefore, the remaining nine candidates were further analyzed, and the raw Ct values obtained using RT-qPCR in various samples are shown in a boxplot (Figure 2). In all samples, the Ct values varied from 21.08 (*GAPA* in leaves spritzed with ABA) to 33.00 (*TBP1* in roots stressed by NaCl), and the average Ct values ranged from 25.89 (*GAPA*) to 30.76 (*CBP20*), indicating that these candidate genes present different expression levels under experimental conditions. Intra- and intergroup statistical analyses of Ct values of nine candidates were further conducted, the results demonstrated that significant differences (*p* < 0.05) were always observed, similar to the results in organs. For example, in the cold group, Ct values of *TBP1* showed significant difference (*p* < 0.001) from that of *GAPA* and *eIF2*, but no significant difference from the other candidates. Among groups, Ct values of *TBP1* in the ABA group had significant difference (*p* < 0.001) from that of the cold and heat groups. In organs, Ct values of *TBP1* in roots showed significant difference (*p* < 0.001) from that of the stems and leaves. However, although *GAPA* exhibited a relatively higher expression, expression bias was evident in various organs as the average Ct values were 30.46, 24.65, and 22.90 in the total roots, stems, and leaves, respectively. Moreover, *TBP1*, *EF-1α*, *CBP20*, and *EIF4A1* had a relatively narrower Ct value range than the other genes, implying these genes might be expressed more stably.

### 3.3. Expression Stability of Candidate Reference Genes

To evaluate the stability of the candidate reference genes, the raw Ct values obtained from all samples were analyzed by geNorm, NormFinder, and BestKeeper; and a comprehensive stability ranking was finally generated by RefFinder. In the geNorm analysis performed for each group, all the groups showed a V2/3 value that was more significant than the threshold value of 0.15 (Appendix A), which suggests that it was difficult to find common reference genes for each treatment. It is better to identify suitable reference genes for individual tissues under a specific condition. Hence, we performed expression stability analysis on two sets of data: (1) a group set that combined the data of all organs under a specific condition; (2) data from individual organs under a specific condition. An investigation of all the stresses and stimuli was also performed.

#### 3.3.1. geNorm Analysis

In geNorm analysis, a cut-off M value of 1.5 is recommended for evaluating all genes’ stability. In this study, the M values of the candidate reference genes were all lower than 1.5 except *GAPA* in all groups and *EF-1α* in the PEG group when analyzed by the group set (Figure 3). Still, they were all lower than 1.5 when analyzed by individual organs in each group (Appendix A). As shown in Figure 3, the two most stable reference genes were not identical under different treatments; they were *TBP1* + *eIF2* for the cold and heat groups, *TBP1* + *EIF4A1* for the PEG and tissue groups, *TBP1* + *CBP20* for the NaCl and total groups, *EF-1α* + *TUB1* for the CuSO_4_ group, *EF-1α* + *EIF4A1* for the ABA group, *EIF4A1* + *TUB1* for the MeJA group, and *CBP20* + *EIF4A1* for the SA group, according to their lowest M values. The most stable reference genes varied more when analyzed by organ; they were not identical among organs even under the same condition (Appendix A). Interestingly, *GAPA* was the most unstable gene in all groups, but it was the most stable gene in the NaCl- and MeJA-treated roots, heat-, PEG-, and SA-treated and CK stems, and CK leaves, which suggests that it can be used as a reference gene under specific conditions. Moreover, the appropriate number of reference genes for normalization was investigated using a threshold value of 0.15 (Vn/n + 1). In our study, the V2/3 values in most organs under a specific condition were lower than 0.15, which meant two reference genes met the requirements for normalization. However, two reference genes were insufficient in CK roots, NaCl-treated stems, and heat-, CuSO_4_-, and ABA-treated leaves (Figure 4).

#### 3.3.2. NormFinder Analysis

Similar to geNorm, the raw Ct values were transformed to relative quantitative values prior to NormFinder analysis. As a result, a variation value of each candidate was given, and all genes were ranked [17]. In this study, the most stable genes could be found directly in each organ under different experimental conditions, and the best combination of two genes was recommended when analyzed by the group set (Table 1). In most cases, the most stable gene varied among organs even under the same treatment, but occasionally it was the same. For example, the most stable gene was *TBP1, EF-1α*, and *TUB1* in the roots, stems, and leaves under cold stress, respectively. However, for the ABA and MeJA stimuli, the most stable gene was the same in the roots, stems, and leaves. As for the best combination of two genes in a specific group, they were *CBP20 + eIF2* for the total group, *ACT7 + eIF2* for the PEG group, *GAPB + TUB1* for the CuSO_4_ group, *TBP1 + eIF2* for the cold, heat, and ABA groups, and *EIF4A1 + eIF2* for the NaCl, MeJA, SA, and tissue groups.

#### 3.3.3. BestKeeper Analysis

Unlike geNorm and NormFinder, the raw Ct data could be used directly by BestKeeper. Finally, standard deviations (SDs) and coefficients of variation (CVs) between pairs of genes were obtained to evaluate the stability of reference genes in each experimental group. The most stable reference gene was considered to have the lowest CV and SD, and the SD values should be less than 1.0. As shown in Table 2, CV ± SD values were calculated and ranked by BestKeeper, and the stability of the reference genes decreased gradually from left to right in the table, similar to NormFinder. In the group’s total and tissue groups, there were three genes with an SD value under the cut-off value of 1.0 in each group, but in the other groups, there were three to six genes. Three genes were in the cold group (*TUB1*, *EIF4A1*, and *EF-1α*), four in the heat group (*EIF4A1*, *CBP20*, *TUB1*, and *TBP1*), four in the PEG group (*EF-1α*, *EIF4A1*, *TBP1*, and *ACT7*), five in the CuSO_4_ group (*TUB1*, *EF-1α*, *CBP20*, *TBP1*, and *EIF4A1*), four in the NaCl group (*CBP20*, *TBP1*, *EF-1α*, and *EIF4A1*), six in the ABA group (*TBP1*, *TUB1*, *ACT7*, *EF-1α*, *CBP20*, and *EIF4A1*), six in the MeJA group (*TUB1*, *EF-1α*, *EIF4A1*, *ACT7*, *CBP20*, and *TBP1*), and five in the SA group (*ACT7*, *TUB1*, *TBP1*, *EF-1α*, and *EIF4A1*). Due to the different analytical principles, the most stable reference gene evaluated by BestKeeper differed somewhat from the results of geNorm and NormFinder. However, the top three most stable reference genes in each group from BestKeeper also had some relatively consistent rankings with geNorm and Normfinder; in particular, *TBP1*, *CBP20*, and *EIF4A1* appeared with the highest frequencies. Unexpectedly, the top two most unstable genes analyzed by the three software programs were the same in all group sets, namely, *GAPA* and *GAPB*. As for the results analyzed with BestKeeper by individual organs in each group, 55.56% (5/9) to 100.00% (9/9) of the candidate reference genes in each organ had an SD value below 1.0.

#### 3.3.4. RefFinder Analysis

Due to the discrepancy in the stability of the candidate reference genes calculated by geNorm, Normfinder, and BestKeeper, the RefFinder program was applied to integrate these results to obtain a comprehensive ranking (Table 3). The ranking order of the top three most stable and unstable reference genes obtained by RefFinder was broadly consistent with the results provided by geNorm and NormFinder but had slight differences with the results from BestKeeper. For instance, under all given experimental conditions, the top two most unstable genes generated by RefFinder appeared in the top three least stable genes yielded by geNorm, Normfinder, and BestKeeper. Ranking orders of the stability of nine reference genes were summarized to better observe the rankings of the analysis results from the four software programs (Appendix A).

### 3.4. Validation of Selected Reference Genes

In this study, *PYL1* exhibited an average Ct value of 34.03 in all samples, which meant a relatively lower expression abundance compared with *ACO1* (33.00) and *CSD2* (26.34). To obtain a more reliable validation result, only the expression analysis of *ACO1* and *CSD2* was conducted. As depicted in Figure 5a, the expression levels of *ACO1* and *CSD2* in the roots and leaves under cold stress were overestimated when the least stable candidate gene was used as the internal control instead of the most stable gene. In contrast, their expression levels in the stems were underestimated, and the normalization differences between the most stable and unstable genes were significant. Moreover, in ABA-treated roots, *ACO1* was enhanced at a similar level (no significant difference) when the two most stable reference genes (*EF-1α* and *ACT7*) and their combination (*EF-1α* + *ACT7*) were used as normalization factors, while a significant difference (*p* < 0.001) was observed when the least stable gene *GAPA* was used (Figure 5b). Therefore, incorrect results would be obtained when using an improper reference gene for the normalization of target genes, highlighting the importance of validating reference genes before conducting an RT-qPCR experiment, to obtain accurate results.

## 4. Discussion

*G. microphylla* is a member of the Fabaceae family. Its seeds are rich in galactomannan and proteins, which are broadly used in industries and animal feed. In addition, the plant has biological characteristics such as drought resistance, barren tolerance, and good adaptability which endows it with good ecological functions. Previous studies [23,24,25,26,27] mainly focused on the application of seeds and the investigation of biological traits, but not on the gene expression and underlying molecular mechanisms of biological processes. Therefore, de novo transcriptome sequencing of *G. microphylla* was conducted by our team for studies concerning sex determination, seed development, stress tolerance, and related gene functions. To achieve these objectives, stable and suitable internal reference genes for RT-qPCR analysis should be identified and evaluated first.

An ideal reference gene should be stably expressed regardless of the experimental condition and cell type. However, no universal reference genes for normalization using RT-qPCR have been discovered, making it particularly important to find proper reference genes when working with tissues of different histological origin or under different conditions. Owning to the absence of reports on reference genes in *G. microphylla*, twelve housekeeping genes (*TBP1*, *CBP20*, *TIP41*, *EF-1α*, *EIF4A1*, *PTB1*, *ACT1*, *ACT7*, *GAPA*, *GAPB*, *TUB1*, and *eIF2*) were selected from our transcriptome datasets for evaluation according to the literature [20,34,35,36,37,38,39]. Among the designed primer pairs for all the candidate reference genes, ten presented high specificity and PCR efficiency, indicating that they could be used in RT-qPCR analysis (Appendix A); and two, *TIP41* (86.90%) and *ACT1* (87.90%), were not further analyzed until higher-efficiency primer pairs were redesigned. In our study, non-coding genes such as rRNAs or miRNAs were not taken into consideration for controversial reasons, because some researchers thought that their high abundance compared with target mRNA transcripts made it difficult to normalize the expression of genes with low expression levels [5,13]. However, other researchers proved they were the best reference genes [7,40,41]. The remaining candidates showed a relatively wide range of expression profiles under the given experimental conditions, confirming again that no single gene could be used for normalization in all the samples of different tissues and under various experimental conditions, similar to the results in *Osmanthus fragrans* [42], *Urochloa brizantha* [43], *Lycoris aurea* [44], and *Eleusine coracana* [45].

The raw Ct values obtained using RT-qPCR in all samples under different conditions were the original data analyzed by the four most popular computational programs (geNorm, NormFinder, BestKeeper, and RefFinder) to rank the reference genes. Therefore, it is vital to ensure an optimal Ct value range for each reference gene through moderate dilution of cDNA. Compared with the other genes, *PTB1*, exhibiting the highest average Ct value of 32.08 and a wider range, was unsuitable as a reference gene. Finally, nine candidate reference genes were evaluated with the four algorithms in this study, and their mean Ct values (Figure 3) and Ct value ranges were sufficient for experimental needs (Appendix A). The stability of genes may be directly reflected in their Ct ranges. For instance, the expression of *TBP1* might be more stable than others according to the relatively narrower Ct range. Importantly, these results were consistent with the outputs calculated by geNorm, NormFinder, BestKeeper, and RefFinder (Figure 3 and Table 1, Table 2 and Table 3). Moreover, these results also reveal that none of the nine candidates could be expressed constantly in all tissues and under different conditions in *G. microphylla*. 

As reference gene determination programs, geNorm evaluated the stability of reference genes with the average pairwise variation [16], and NormFinder exhibited the expression stability of reference genes by analyzing their intra- and intergroup variation [17]. However, the CV and SD values were the key factors in determining the stability rankings of the reference genes obtained by BestKeeper [18]. Thus, the ranking results were not identical and were reasonable. In this study, *TBP1* and *EIF4A1* were the most stable genes, and *GAPA* and *GAPB* were the most unstable genes under different conditions in geNorm analysis, which was relatively consistent with the results returned by NormFinder, and thus further ensured the accuracy and reliability of the analysis results. However, *EF-1α*, the most unstable reference gene in the PEG group from geNorm and NormFinder analyses, was ranked at a top position in the BestKeeper analysis. Similar research findings were reported in the literature on the selection and validation of reference genes in other plant species such as *R. yunnanensis* [20] and *sorghum bicolor* [46]. Fortunately, BestKeeper still showed some conformance to the top three most stable reference genes with geNorm and Normfinder (Figure 4; Table 2 and Table 3). Finally, a widely used web-based tool, RefFinder, was used to obtain a consensus stability ranking of each candidate gene under the given conditions according to the geometric mean of the attributed weights of each gene [19]. For the RefFinder analysis, *TBP1*, *EIF4A1*, and *CBP20* were ranked as the top three most stable reference genes, similar to geNorm, Normfinder, and BestKeeper. Synthetically, *TBP1*, *EIF4A1*, *CBP20*, and *TUB1* were the four most frequent and stable genes ranked by the four programs, with *TBP1* occurring 38 times, *EIF4A1* 36 times, *CBP20* 25 times, and *TUB1* 21 times. Among them, the top four most stable genes returned by BestKeeper were *TBP1* nine times, *EIF4A1* eight times, *CBP20* six times, and *TUB1* six times. Correspondingly, the top four most unstable genes in all groups were *GAPA* occurring 40 times, *GAPB* occurring 39 times, *eIF2* occurring 23 times, and *EF-1α* occurring 22 times. The most unstable gene seemed to be determined from the top four most unstable genes in each group (Appendix A).

According to the stability value, *TBP1*, *EIF4A1*, and *CBP20* were considered as the most stable internal reference genes in all samples. However, the most stable reference genes were not identical in different groups; they differed even among organs in the same group. Therefore, we suggest that it is better to choose the best reference genes for a specific case rather than using a common one for normalization, although this is slightly laborious and time-consuming. Theoretically, the proteins encoded by housekeeping genes are used either to maintain the cell structure or participate in basic cellular metabolic processes, and they should be stably expressed regardless of the tissue type or the physiological state. However, some housekeeping genes have been proven to have poor expression stability in specific experimental conditions and could not act as internal reference genes for RT-qPCR analysis [7,9]. In this study, the traditional housekeeping genes *TBP1*, *EIF4A1*, and *CBP20* exhibited reasonably good stability under the given conditions. To obtain a more accurate normalization result, an increasing number of studies have applied multiple reference genes rather than a single reference gene [46,47].

In response to environmental stimuli, the expression level of *ACO1* (encoding ethylene-forming enzyme) and *CSD2* (encoding superoxide radicals detoxifying enzyme) would change in a wide range in *G. microphylla.* Thus, these two stress-responding genes were chosen to validate the suitability of the reference genes selected in our study. The most stable and unstable reference genes recommended by RefFinder in the roots, stems, and leaves under cold treatment were used to normalize *ACO1* and *CSD2*. The relative expression levels of *ACO1* and *CSD2* involved in stress responses had significant differences when normalized with the most stable and unstable genes (Figure 5a). In ABA treated roots, the relative expression levels of *ACO1* showed a similar level (no significant difference) when normalized with the two most stable genes alone or in combination. Still, a lower expression level (significant difference) was observed when normalized with the least stable gene, *GAPA* (Figure 5b), suggesting that the selection and confirmation of suitable and stable reference genes were particularly critical for the proper normalization for the RT-qPCR data in *G. microphylla*.

## 5. Conclusions

Our study evaluated the expression stability of nine candidate reference genes selected from transcriptome data of *G. microphylla* under a wide range of experimental conditions, five types of abiotic stress (cold, heat, drought, salinity, and heavy-metal) and three types of hormone stimulus (MeJA, ABA, and SA). According to the comprehensive results analyzed by four widely used programs (geNorm, NormFinder, BestKeeper, and RefFinder), *TBP1* and *EIF4A1* were the most stable genes, and *GAPA* and *GAPB* were the most unstable genes, in all groups. Meanwhile, the most stable genes varied among different conditions and tissues (root, stem, leaf, and flower), suggesting that it is better to choose the best reference gene for a specific case rather than using a common one, and normalization with two- or multi-gene combinations is encouraged. In addition, the expression analysis of *ACO1* and *CSD2* emphasized the importance of selecting suitable and stable genes for the normalization of gene expression analysis using RT-qPCR. This study is the first report on the selection and validation of reference genes in *G. microphylla* and related species of *Gleditsia*. It provides an essential foundation for future research on gene expression analyses using RT-qPCR. 

## Figures and Tables

**Figure 1 genes-13-01227-f001:**
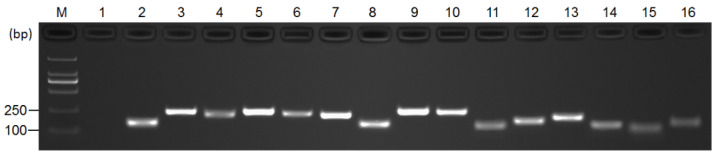
Specificity of primer pairs for RT-qPCR amplification of twelve candidate reference genes and three target genes. The 2% agarose gel electrophoresis shows the expected size of a single band of each candidate reference gene and target gene. M: 2K DNA marker; Lane 1: no-template control; Lane 2 to Lane 16: *TBP1*, *ACT7*, *CBP20*, *EF-1α*, *EIF4A1*, *GAPA*, *GAPB*, *PTB1*, *TIP41*, *ACT1*, *eIF2*, *TUB1*, *ACO1*, *PYL1* and *CSD2*.

**Figure 2 genes-13-01227-f002:**
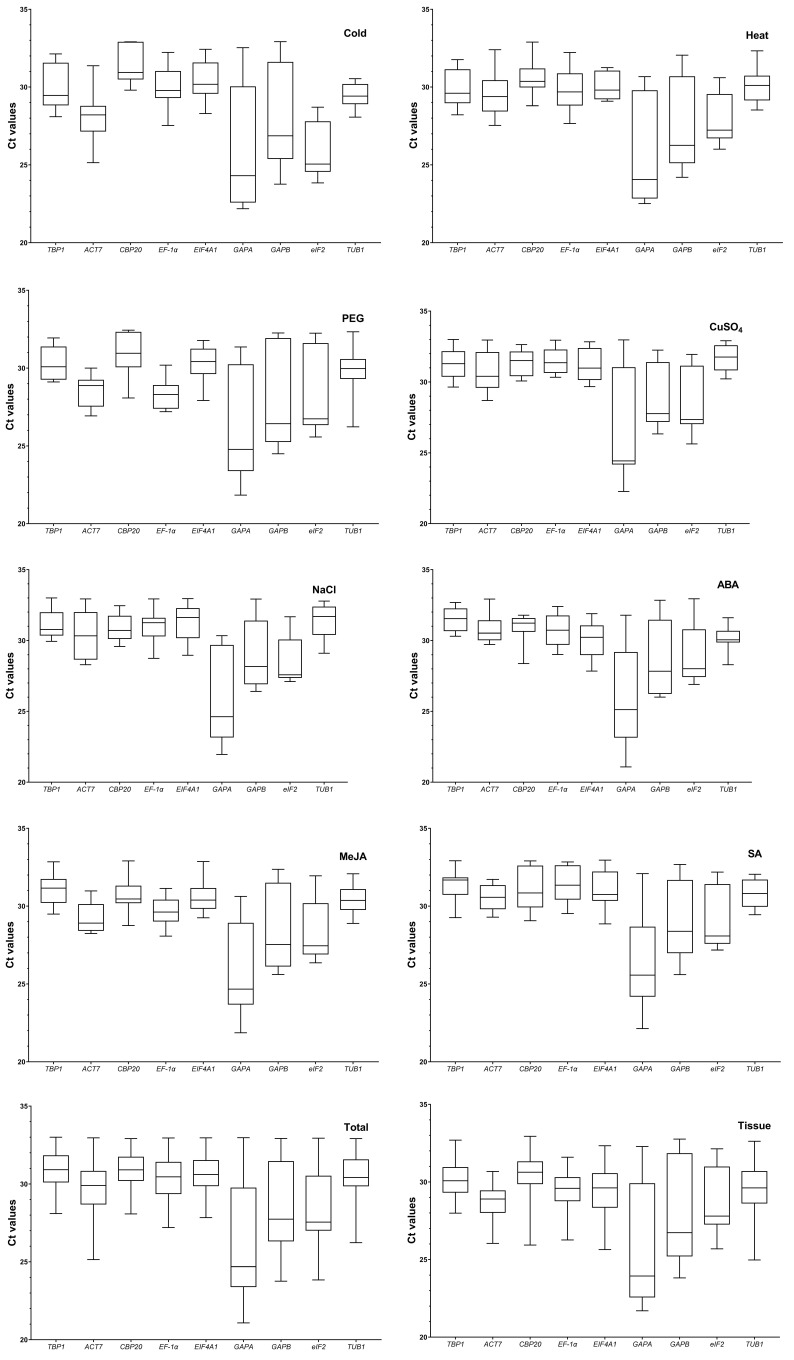
RT-qPCR raw Ct values for nine candidate reference genes in different groups. The box indicates the 25th and 75th percentiles, and whisker caps show the maximum and minimum values. The line across the box depicts the median; the lower the boxes and whisker, the smaller the variations. Cold and Heat represent different temperature shock groups; PEG, CuSO_4_, and NaCl represent drought, heavy-metal, and salinity stress groups, respectively; ABA, MeJA, and SA represent different hormone stimulus groups; Total represents a group of all the samples under experimental conditions; Tissue represents a group of samples including roots, stems and leaves from various control groups and flowers.

**Figure 3 genes-13-01227-f003:**
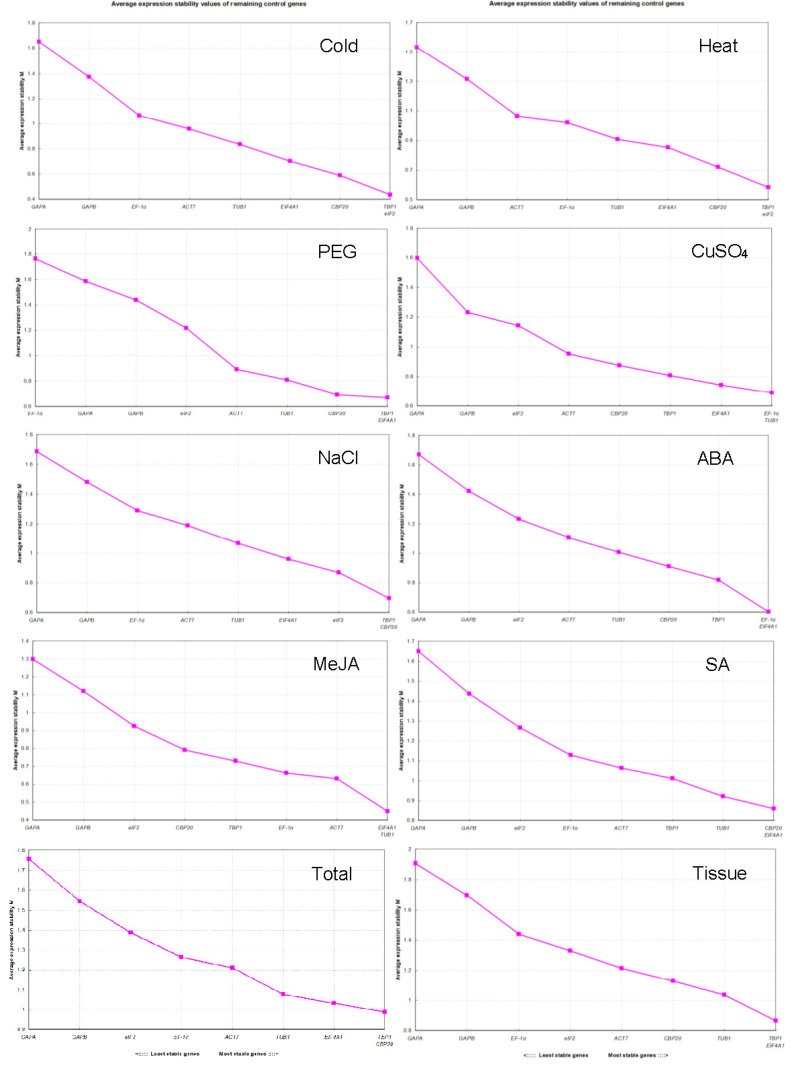
Average expression stability values M of nine candidate reference genes calculated by geNorm with samples from given experimental conditions. The most unstable genes are on the left and the most stable genes are on the right. Cold, Heat, PEG, CuSO_4_, and NaCl represent samples from different abiotic stresses; ABA, MeJA, and SA represent samples from different hormone treatments; Tissue represents samples from roots, stems, and leaves of CK and flowers; Total represents samples from all abiotic stress and hormone treatment groups.

**Figure 4 genes-13-01227-f004:**
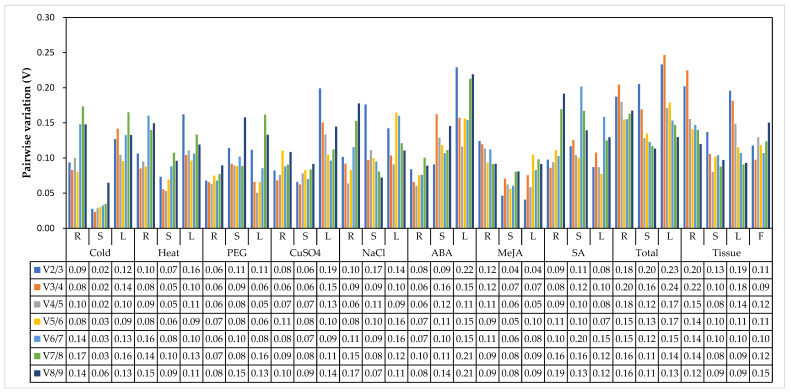
The pairwise variation (V) measures of the nine candidate reference genes using geNorm. Vn/n + 1 values were used to calculate the optimal number of reference genes (n). The letters R, S, L, and F represent root, stem, leaf, and flower, respectively.

**Figure 5 genes-13-01227-f005:**
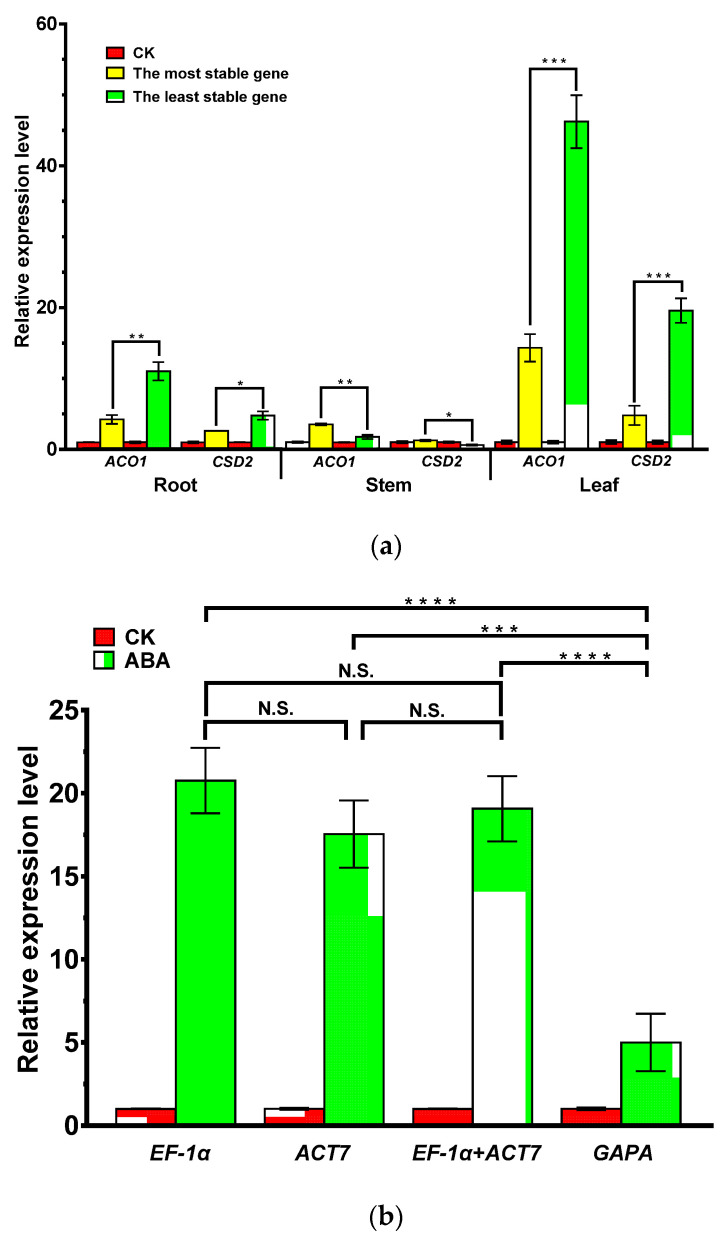
Validation of the selected reference genes. Relative expression levels of *ACO1* and *CSD2* were normalized using candidate reference genes under different treatments. (**a**) The expression level was normalized using the most stable and unstable reference genes in various organs under cold treatment. The most stable and unstable reference genes in the roots, stems, and leaves were *TBP1* and *EF-1α*, *EF-1α* and *ACT7*, and *TUB1* and *EF-1α*, respectively. (**b**) The expression level of *ACO1* was normalized using the most stable reference genes and their combination in roots under ABA treatment. *EF-1α* and *ACT7* represent the two most stable reference genes, and *GAPA* is the most unstable gene. Data are displayed as means ± SEM (*n* = 3), and the statistical analyses were performed using multiple comparisons of one-way ANOVA to compare those between two reference genes or combinations for normalization. * *p* < 0.05; ** *p* < 0.01; *** *p* < 0.001, **** *p* < 0.0001. N.S.: no significant difference.

**Table 1 genes-13-01227-t001:** Expression stability values for nine candidate reference genes calculated by NormFinder.

Group	Organ	Rank									Best Combination
		1	2	3	4	5	6	7	8	9	
Total	RSL	*TBP1*(0.342)	*EIF4A1*(0.422)	*eIF2*(0.462)	*ACT7*(0.493)	*CBP20*(0.501)	*TUB1*(0.618)	*EF-1α*(0.663)	*GAPB*(0.825)	*GAPA*(1.301)	*CBP20/eIF2*
	R	*TBP1*(0.257)	*EIF4A1*(0.299)	*GAPB*(0.351)	*TUB1*(0.486)	*GAPA*(0.630)	*eIF2*(0.707)	*ACT7*(0.761)	*CBP20*(0.773)	*EF-1α*(1.024)	
	S	*GAPB*(0.261)	*EIF4A1*(0.369)	*TUB1*(0.402)	*TBP1*(0.419)	*GAPA*(0.503)	*CBP20*(0.515)	*eIF2*(0.543)	*EF-1α*(0.571)	*ACT7*(0.688)	
	L	*GAPB*(0.210)	*eIF2*(0.442)	*TUB1*(0.485)	*TBP1*(0.497)	*EIF4A1*(0.643)	*CBP20*(0.707)	*EF-1α*(0.743)	*GAPA*(0.760)	*ACT7*(0.781)	
Cold	RSL	*eIF2*(0.188)	*TBP1*(0.214)	*EIF4A1*(0.453)	*CBP20*(0.494)	*ACT7*(0.636)	*TUB1*(0.724)	*EF-1α*(1.027)	*GAPB*(1.065)	*GAPA*(1.489)	*TBP1/eIF2*
	R	*TBP1*(0.051)	*TUB1*(0.051)	*eIF2*(0.064)	*GAPB*(0.173)	*CBP20*(0.315)	*EIF4A1*(0.435)	*ACT7*(0.814)	*GAPA*(0.898)	*EF-1α*(0.908)	
	S	*EF-1α*(0.094)	*CBP20*(0.096)	*TUB1*(0.099)	*eIF2*(0.100)	*EIF4A1*(0.103)	*TBP1*(0.109)	*GAPA*(0.126)	*GAPB*(0.127)	*ACT7*(0.402)	
	L	*TUB1*(0.110)	*GAPB*(0.114)	*EIF4A1*(0.166)	*eIF2*(0.220)	*GAPA*(0.417)	*TBP1*(0.437)	*ACT7*(0.787)	*EF-1α*(0.811)	*CBP20*(0.828)	
Heat	RSL	*eIF2*(0.161)	*TBP1*(0.393)	*CBP20*(0.606)	*TUB1*(0.664)	*EIF4A1*(0.674)	*EF-1α*(0.684)	*ACT7*(0.711)	*GAPB*(0.937)	*GAPA*(1.300)	*TBP1/eIF2*
	R	*GAPA*(0.047)	*eIF2*(0.047)	*GAPB*(0.077)	*TUB1*(0.184)	*EIF4A1*(0.328)	*TBP1*(0.354)	*EF-1α*(0.749)	*ACT7*(0.860)	*CBP20*(0.920)	
	S	*eIF2*(0.063)	*TBP1*(0.101)	*GAPA*(0.214)	*CBP20*(0.251)	*GAPB*(0.255)	*EIF4A1*(0.331)	*TUB1*(0.385)	*EF-1α*(0.505)	*ACT7*(0.588)	
	L	*TUB1*(0.106)	*TBP1*(0.223)	*eIF2*(0.251)	*EIF4A1*(0.251)	*GAPB*(0.256)	*GAPA*(0.446)	*ACT7*(0.640)	*CBP20*(0.653)	*EF-1α*(0.730)	
PEG	RSL	*EIF4A1*(0.399)	*TBP1*(0.471)	*CBP20*(0.538)	*TUB1*(0.663)	*ACT7*(0.781)	*eIF2*(0.822)	*GAPB*(1.062)	*GAPA*(1.380)	*EF-1α*(1.483)	*ACT7/eIF2*
	R	*GAPB*(0.037)	*EIF4A1*(0.039)	*eIF2*(0.065)	*TBP1*(0.156)	*CBP20*(0.268)	*ACT7*(0.272)	*GAPA*(0.277)	*EF-1α*(0.474)	*TUB1*(0.550)	
	S	*EIF4A1*(0.066)	*eIF2*(0.097)	*GAPB*(0.227)	*ACT7*(0.242)	*TBP1*(0.325)	*TUB1*(0.342)	*CBP20*(0.471)	*GAPA*(0.502)	*EF-1α*(0.976)	
	L	*EIF4A1*(0.043)	*TUB1*(0.043)	*GAPB*(0.070)	*eIF2*(0.227)	*TBP1*(0.228)	*GAPA*(0.377)	*CBP20*(0.598)	*ACT7*(0.768)	*EF-1α*(0.818)	
CuSO_4_	RSL	*TBP1*(0.405)	*ACT7*(0.406)	*EF-1α*(0.511)	*GAPB*(0.561)	*eIF2*(0.609)	*EIF4A1*(0.657)	*TUB1*(0.699)	*CBP20*(0.917)	*GAPA*(1.677)	*GAPB/TUB1*
	R	*TBP1*(0.054)	*eIF2*(0.078)	*GAPB*(0.086)	*EF-1α*(0.139)	*EIF4A1*(0.232)	*TUB1*(0.417)	*ACT7*(0.482)	*GAPA*(0.513)	*CBP20*(0.668)	
	S	*GAPA*(0.071)	*GAPB*(0.120)	*eIF2*(0.165)	*TBP1*(0.182)	*EIF4A1*(0.265)	*TUB1*(0.286)	*EF-1α*(0.373)	*CBP20*(0.433)	*ACT7*(0.562)	
	L	*CBP20*(0.206)	*EF-1α*(0.269)	*eIF2*(0.319)	*TBP1*(0.343)	*ACT7*(0.343)	*TUB1*(0.390)	*GAPB*(0.397)	*GAPA*(0.705)	*EIF4A1*(0.891)	
NaCl	RSL	*TBP1*(0.258)	*eIF2*(0.307)	*EIF4A1*(0.460)	*CBP20*(0.611)	*ACT7*(0.714)	*TUB1*(0.739)	*GAPB*(0.962)	*EF-1α*(1.116)	*GAPA*(1.361)	*EIF4A1/eIF2*
	R	*EIF4A1*(0.071)	*eIF2*(0.071)	*GAPB*(0.181)	*TUB1*(0.182)	*GAPA*(0.232)	*TBP1*(0.252)	*EF-1α*(0.498)	*CBP20*(0.937)	*ACT7*(1.099)	
	S	*TBP1*(0.201)	*eIF2*(0.205)	*EF-1α*(0.274)	*GAPB*(0.282)	*CBP20*(0.367)	*GAPA*(0.403)	*ACT7*(0.410)	*EIF4A1*(0.417)	*TUB1*(0.434)	
	L	*eIF2*(0.133)	*TBP1*(0.266)	*GAPB*(0.311)	*TUB1*(0.346)	*GAPA*(0.529)	*EIF4A1*(0.593)	*CBP20*(0.617)	*EF-1α*(0.638)	*ACT7*(0.669)	
ABA	RSL	*EF-1α*(0.237)	*EIF4A1*(0.346)	*eIF2*(0.547)	*TBP1*(0.576)	*CBP20*(0.627)	*TUB1*(0.815)	*GAPB*(0.860)	*ACT7*(0.897)	*GAPA*(1.408)	*TBP1/eIF2*
	R	*EF-1α*(0.039)	*EIF4A1*(0.048)	*ACT7*(0.051)	*TBP1*(0.205)	*GAPB*(0.224)	*CBP20*(0.301)	*eIF2*(0.468)	*TUB1*(0.476)	*GAPA*(0.546)	
	S	*EF-1α*(0.136)	*GAPB*(0.137)	*eIF2*(0.296)	*TUB1*(0.359)	*TBP1*(0.360)	*EIF4A1*(0.371)	*CBP20*(0.498)	*GAPA*(0.701)	*ACT7*(0.896)	
	L	*EF-1α*(0.093)	*GAPB*(0.093)	*eIF2*(0.139)	*TBP1*(0.340)	*EIF4A1*(0.631)	*TUB1*(0.644)	*CBP20*(0.995)	*ACT7*(0.996)	*GAPA*(1.351)	
MeJA	RSL	*TBP1*(0.358)	*eIF2*(0.417)	*EIF4A1*(0.444)	*ACT7*(0.512)	*CBP20*(0.543)	*TUB1*(0.554)	*EF-1α*(0.611)	*GAPB*(0.774)	*GAPA*(1.083)	*EIF4A1/eIF2*
	R	*TUB1*(0.133)	*GAPB*(0.167)	*eIF2*(0.253)	*EIF4A1*(0.351)	*TBP1*(0.372)	*GAPA*(0.427)	*EF-1α*(0.471)	*ACT7*(0.496)	*CBP20*(0.557)	
	S	*TUB1*(0.019)	*GAPB*(0.046)	*eIF2*(0.068)	*GAPA*(0.203)	*TBP1*(0.262)	*EIF4A1*(0.274)	*CBP20*(0.276)	*EF-1α*(0.400)	*ACT7*(0.497)	
	L	*TUB1*(0.040)	*GAPB*(0.090)	*EIF4A1*(0.097)	*TBP1*(0.149)	*eIF2*(0.176)	*ACT7*(0.453)	*CBP20*(0.456)	*GAPA*(0.499)	*EF-1α*(0.559)	
SA	RSL	*EIF4A1*(0.476)	*eIF2*(0.520)	*CBP20*(0.526)	*TBP1*(0.602)	*TUB1*(0.615)	*ACT7*(0.657)	*EF-1α*(0.687)	*GAPB*(0.831)	*GAPA*(1.326)	*EIF4A1/eIF2*
	R	*eIF2*(0.086)	*GAPB*(0.103)	*TUB1*(0.194)	*EIF4A1*(0.219)	*TBP1*(0.283)	*CBP20*(0.541)	*GAPA*(0.729)	*ACT7*(0.753)	*EF-1α*(1.184)	
	S	*EIF4A1*(0.085)	*TUB1*(0.085)	*CBP20*(0.318)	*GAPB*(0.442)	*GAPA*(0.560)	*ACT7*(0.724)	*eIF2*(0.730)	*EF-1α*(0.806)	*TBP1*(0.848)	
	L	*ACT7*(0.053)	*eIF2*(0.053)	*GAPB*(0.126)	*GAPA*(0.359)	*CBP20*(0.462)	*TBP1*(0.553)	*EIF4A1*(0.596)	*EF-1α*(0.612)	*TUB1*(0.793)	
Tissue	RSLF	*TBP1*(0.378)	*EIF4A1*(0.394)	*eIF2*(0.482)	*TUB1*(0.676)	*ACT7*(0.730)	*CBP20*(0.877)	*GAPB*(1.001)	*EF-1α*(1.092)	*GAPA*(1.291)	*EIF4A1/eIF2*
	R	*TBP1*(0.210)	*GAPB*(0.331)	*EIF4A1*(0.434)	*ACT7*(0.466)	*TUB1*(0.536)	*GAPA*(0.587)	*EF-1α*(0.714)	*CBP20*(0.723)	*eIF2*(0.724)	
	S	*EIF4A1*(0.183)	*eIF2*(0.222)	*TBP1*(0.232)	*GAPA*(0.269)	*CBP20*(0.367)	*GAPB*(0.394)	*ACT7*(0.421)	*EF-1α*(0.454)	*TUB1*(0.594)	
	L	*TUB1*(0.312)	*EIF4A1*(0.319)	*eIF2*(0.376)	*EF-1α*(0.408)	*TBP1*(0.440)	*GAPB*(0.440)	*CBP20*(0.488)	*GAPA*(0.490)	*ACT7*(0.560)	
	F	*GAPB*(0.135)	*TBP1*(0.163)	*EIF4A1*(0.169)	*ACT7*(0.323)	*TUB1*(0.331)	*CBP20*(0.460)	*eIF2*(0.472)	*GAPA*(0.756)	*EF-1α*(0.927)	

**Table 2 genes-13-01227-t002:** Expression stability values for nine candidate reference genes calculated by BestKeeper.

Group	Organ	Rank *								
		1	2	3	4	5	6	7	8	9
Total	RSL	*CBP20*	*EIF4A1*	*TUB1*	*TBP1*	*EF-1α*	*ACT7*	*eIF2*	*GAPB*	*GAPA*
		3.00 ± 0.93	3.08 ± 0.94	3.16 ± 0.96	3.27 ± 1.01	3.92 ± 1.19	4.12 ± 1.23	6.41 ± 1.81	8.42 ± 2.40	11.38 ± 2.96
	R	*TBP1*	*GAPB*	*EIF4A1*	*CBP20*	*TUB1*	*GAPA*	*eIF2*	*ACT7*	*EF-1α*
		1.48 ± 0.47	1.54 ± 0.49	1.59 ± 0.50	2.27 ± 0.72	2.53 ± 0.79	2.80 ± 0.85	3.27 ± 1.01	3.95 ± 1.21	4.60 ± 1.42
	S	*TBP1*	*CBP20*	*EIF4A1*	*TUB1*	*GAPB*	*GAPA*	*EF-1α*	*eIF2*	*ACT7*
		2.28 ± 0.70	2.39 ± 0.74	2.73 ± 0.82	2.90 ± 0.86	2.94 ± 0.81	3.03 ± 0.75	3.08 ± 0.93	3.16 ± 0.86	3.49 ± 1.02
	L	*CBP20*	*EIF4A1*	*TUB1*	*GAPB*	*TBP1*	*GAPA*	*eIF2*	*EF-1α*	*ACT7*
		2.19 ± 0.66	2.33 ± 0.70	2.57 ± 0.79	2.63 ± 0.68	2.92 ± 0.87	2.99 ± 0.69	3.07 ± 0.82	3.63 ± 1.09	3.90 ± 1.15
Cold	RSL	*TUB1*	*CBP20*	*EIF4A1*	*EF-1α*	*ACT7*	*TBP1*	*eIF2*	*GAPB*	*GAPA*
		2.05 ± 0.60	3.23 ± 1.01	3.24 ± 0.99	3.29 ± 0.98	4.06 ± 1.14	4.24 ± 1.27	6.06 ± 1.57	10.12 ± 2.83	12.77 ± 3.30
	R	*CBP20*	*TUB1*	*TBP1*	*EIF4A1*	*eIF2*	*GAPB*	*GAPA*	*EF-1α*	*ACT7*
		0.02 ± 0.01	0.91 ± 0.27	0.91 ± 0.29	1.11 ± 0.35	1.74 ± 0.49	1.83 ± 0.59	3.48 ± 1.07	3.90 ± 1.19	3.92 ± 1.16
	S	*CBP20*	*EIF4A1*	*TBP1*	*eIF2*	*TUB1*	*GAPB*	*EF-1α*	*GAPA*	*ACT7*
		0.09 ± 0.03	0.45 ± 0.13	0.47 ± 0.14	0.55 ± 0.14	0.65 ± 0.19	0.74 ± 0.20	0.62 ± 0.18	0.92 ± 0.22	1.53 ± 0.42
	L	*GAPA*	*CBP20*	*eIF2*	*TBP1*	*TUB1*	*GAPB*	*EIF4A1*	*EF-1α*	*ACT7*
		0.65 ± 0.15	0.99 ± 0.30	1.53 ± 0.38	1.86 ± 0.53	2.38 ± 0.70	2.85 ± 0.71	3.09 ± 0.92	5.20 ± 1.55	5.33 ± 1.46
Heat	RSL	*EIF4A1*	*CBP20*	*TUB1*	*TBP1*	*EF-1α*	*ACT7*	*eIF2*	*GAPB*	*GAPA*
		2.52 ± 0.76	2.74 ± 0.84	2.91 ± 0.87	3.29 ± 0.98	3.70 ± 1.10	4.01 ± 1.18	5.04 ± 1.40	8.97 ± 2.47	11.65 ± 2.99
	R	*EIF4A1*	*TBP1*	*GAPA*	*eIF2*	*GAPB*	*CBP20*	*TUB1*	*EF-1α*	*ACT7*
		0.56 ± 0.17	0.76 ± 0.24	1.21 ± 0.36	1.32 ± 0.40	1.81 ± 0.57	1.93 ± 0.61	2.41 ± 0.75	3.99 ± 1.24	4.40 ± 1.35
	S	*GAPA*	*TBP1*	*EIF4A1*	*eIF2*	*TUB1*	*CBP20*	*GAPB*	*EF-1α*	*ACT7*
		0.52 ± 0.13	0.52 ± 0.16	0.65 ± 0.19	0.68 ± 0.18	0.72 ± 0.21	0.80 ± 0.24	1.99 ± 0.53	2.33 ± 0.68	2.84 ± 0.82
	L	*GAPA*	*TBP1*	*TUB1*	*GAPB*	*eIF2*	*EIF4A1*	*CBP20*	*ACT7*	*EF-1α*
		0.73 ± 0.17	1.08 ± 0.31	1.31 ± 0.39	1.33 ± 0.33	1.74 ± 0.46	2.23 ± 0.67	2.36 ± 0.70	3.10 ± 0.90	3.61 ± 1.06
PEG	RSL	*EF-1α*	*EIF4A1*	*TBP1*	*ACT7*	*TUB1*	*CBP20*	*eIF2*	*GAPB*	*GAPA*
		2.64 ± 0.75	2.88 ± 0.87	2.91 ± 0.88	3.07 ± 0.88	3.37 ± 1.01	3.67 ± 1.14	8.28 ± 2.35	9.78 ± 2.73	11.34 ± 2.98
	R	*CBP20*	*EIF4A1*	*GAPB*	*TBP1*	*eIF2*	*EF-1α*	*ACT7*	*GAPA*	*TUB1*
		0.41 ± 0.13	0.41 ± 0.13	0.43 ± 0.14	0.69 ± 0.22	0.93 ± 0.30	1.12 ± 0.31	1.20 ± 0.35	1.29 ± 0.40	2.10 ± 0.66
	S	*EF-1α*	*TBP1*	*ACT7*	*EIF4A1*	*eIF2*	*GAPB*	*TUB1*	*CBP20*	*GAPA*
		0.48 ± 0.14	1.79 ± 0.54	2.06 ± 0.58	3.23 ± 0.95	3.25 ± 0.87	4.20 ± 1.11	4.56 ± 1.30	4.59 ± 1.40	5.90 ± 1.44
	L	*eIF2*	*TBP1*	*TUB1*	*EIF4A1*	*GAPB*	*GAPA*	*CBP20*	*ACT7*	*EF-1α*
		0.14 ± 0.04	0.29 ± 0.08	0.73 ± 0.22	0.86 ± 0.26	0.97 ± 0.25	1.09 ± 0.26	1.24 ± 0.37	3.96 ± 1.12	4.17 ± 1.18
CuSO_4_	RSL	*TUB1*	*EF-1α*	*CBP20*	*TBP1*	*EIF4A1*	*ACT7*	*GAPB*	*eIF2*	*GAPA*
		2.22 ± 0.70	2.29 ± 0.72	2.39 ± 0.75	2.76 ± 0.86	2.87 ± 0.89	3.73 ± 1.14	6.76 ± 1.94	7.11 ± 2.02	13.10 ± 3.47
	R	*EF-1α*	*GAPB*	*TBP1*	*ACT7*	*TUB1*	*eIF2*	*EIF4A1*	*GAPA*	*CBP20*
		0.36 ± 0.12	0.55 ± 0.17	0.87 ± 0.28	1.04 ± 0.34	1.10 ± 0.36	1.16 ± 0.36	1.55 ± 0.50	2.16 ± 0.69	2.88 ± 0.91
	S	*GAPB*	*EIF4A1*	*eIF2*	*GAPA*	*EF-1α*	*TBP1*	*TUB1*	*CBP20*	*ACT7*
		0.02 ± 0.01	0.56 ± 0.17	0.57 ± 0.15	0.93 ± 0.23	1.27 ± 0.39	1.32 ± 0.41	1.66 ± 0.51	1.87 ± 0.58	2.05 ± 0.61
	L	*EF-1α*	*GAPB*	*ACT7*	*TUB1*	*TBP1*	*CBP20*	*eIF2*	*GAPA*	*EIF4A1*
		1.08 ± 0.33	1.16 ± 0.31	1.26 ± 0.38	1.76 ± 0.56	1.83 ± 0.56	2.13 ± 0.66	2.87 ± 0.77	2.97 ± 0.69	3.72 ± 1.14
NaCl	RSL	*CBP20*	*TBP1*	*EF-1α*	*EIF4A1*	*TUB1*	*ACT7*	*eIF2*	*GAPB*	*GAPA*
		2.47 ± 0.76	2.51 ± 0.78	2.79 ± 0.87	2.93 ± 0.92	3.23 ± 1.01	4.11 ± 1.25	4.85 ± 1.39	7.50 ± 2.18	10.43 ± 2.69
	R	*GAPA*	*TUB1*	*TBP1*	*EIF4A1*	*eIF2*	*GAPB*	*CBP20*	*EF-1α*	*ACT7*
		0.85 ± 0.25	1.08 ± 0.35	1.26 ± 0.40	1.51 ± 0.48	1.84 ± 0.56	2.22 ± 0.71	2.55 ± 0.81	3.58 ± 1.09	5.85 ± 1.83
	S	*GAPA*	*TBP1*	*TUB1*	*CBP20*	*GAPB*	*ACT7*	*EF-1α*	*eIF2*	*EIF4A1*
		0.56 ± 0.14	1.12 ± 0.34	1.73 ± 0.52	1.78 ± 0.55	1.79 ± 0.50	2.41 ± 0.71	2.52 ± 0.78	2.73 ± 0.76	3.19 ± 0.99
	L	*eIF2*	*TBP1*	*GAPB*	*CBP20*	*GAPA*	*TUB1*	*ACT7*	*EF-1α*	*EIF4A1*
		0.18 ± 0.05	0.62 ± 0.19	0.75 ± 0.20	1.49 ± 0.45	1.56 ± 0.36	2.22 ± 0.71	2.74 ± 0.84	2.87 ± 0.91	3.13 ± 0.97
ABA	RSL	*TBP1*	*TUB1*	*ACT7*	*EF-1α*	*CBP20*	*EIF4A1*	*eIF2*	*GAPB*	*GAPA*
		2.14 ± 0.67	2.23 ± 0.67	2.39 ± 0.74	3.15 ± 0.97	3.21 ± 0.98	3.28 ± 0.99	6.00 ± 1.74	7.93 ± 2.28	10.69 ± 2.78
	R	*CBP20*	*TBP1*	*TUB1*	*EIF4A1*	*EF-1α*	*ACT7*	*GAPB*	*eIF2*	*GAPA*
		0.22 ± 0.07	0.38 ± 0.12	0.56 ± 0.17	0.77 ± 0.24	0.98 ± 0.32	1.36 ± 0.42	1.91 ± 0.61	2.70 ± 0.85	3.02 ± 0.91
	S	*eIF2*	*TBP1*	*ACT7*	*GAPB*	*TUB1*	*EF-1α*	*EIF4A1*	*CBP20*	*GAPA*
		0.96 ± 0.27	1.09 ± 0.34	1.57 ± 0.48	1.64 ± 0.46	1.96 ± 0.57	2.17 ± 0.66	3.42 ± 1.01	3.74 ± 1.13	5.41 ± 1.33
	L	*EF-1α*	*GAPB*	*eIF2*	*EIF4A1*	*TBP1*	*TUB1*	*ACT7*	*CBP20*	*GAPA*
		0.23 ± 0.07	0.29 ± 0.08	1.27 ± 0.35	2.17 ± 0.64	2.17 ± 0.68	2.21 ± 0.67	3.53 ± 1.09	4.02 ± 1.21	5.68 ± 1.33
MeJA	RSL	*TUB1*	*EF-1α*	*EIF4A1*	*ACT7*	*CBP20*	*TBP1*	*eIF2*	*GAPB*	*GAPA*
		2.11 ± 0.64	2.23 ± 0.66	2.43 ± 0.74	2.49 ± 0.73	2.69 ± 0.82	2.80 ± 0.87	5.80 ± 1.64	7.74 ± 2.20	9.69 ± 2.51
	R	*GAPB*	*TUB1*	*EF-1α*	*ACT7*	*EIF4A1*	*eIF2*	*TBP1*	*CBP20*	*GAPA*
		0.48 ± 0.15	0.70 ± 0.22	0.89 ± 0.27	1.16 ± 0.35	1.19 ± 0.38	1.65 ± 0.51	1.77 ± 0.57	2.17 ± 0.69	2.19 ± 0.65
	S	*ACT7*	*eIF2*	*TUB1*	*GAPB*	*EF-1α*	*TBP1*	*CBP20*	*EIF4A1*	*GAPA*
		0.81 ± 0.23	0.83 ± 0.23	1.08 ± 0.32	1.23 ± 0.34	1.50 ± 0.44	1.99 ± 0.61	2.00 ± 0.61	2.03 ± 0.61	2.09 ± 0.52
	L	*TUB1*	*EIF4A1*	*eIF2*	*GAPB*	*TBP1*	*ACT7*	*EF-1α*	*CBP20*	*GAPA*
		0.34 ± 0.10	0.63 ± 0.19	0.75 ± 0.20	0.77 ± 0.20	1.11 ± 0.34	1.31 ± 0.38	1.62 ± 0.47	2.28 ± 0.68	3.12 ± 0.72
SA	RSL	*ACT7*	*TUB1*	*TBP1*	*EF-1α*	*EIF4A1*	*CBP20*	*eIF2*	*GAPB*	*GAPA*
		2.11 ± 0.65	2.22 ± 0.69	2.22 ± 0.70	2.88 ± 0.90	3.15 ± 0.98	3.59 ± 1.12	6.35 ± 1.86	8.05 ± 2.35	10.77 ± 2.86
	R	*GAPB*	*TBP1*	*ACT7*	*eIF2*	*EIF4A1*	*TUB1*	*EF-1α*	*CBP20*	*GAPA*
		1.11 ± 0.36	1.74 ± 0.56	1.77 ± 0.55	1.92 ± 0.61	2.41 ± 0.76	2.73 ± 0.85	3.19 ± 1.02	3.74 ± 1.19	4.23 ± 1.28
	S	*TBP1*	*TUB1*	*EF-1α*	*ACT7*	*EIF4A1*	*CBP20*	*GAPB*	*GAPA*	*eIF2*
		0.96 ± 0.30	2.13 ± 0.66	2.16 ± 0.68	2.45 ± 0.75	2.70 ± 0.85	3.03 ± 0.95	4.18 ± 1.22	4.63 ± 1.21	5.23 ± 1.51
	L	*eIF2*	*ACT7*	*TUB1*	*EF-1α*	*GAPB*	*EIF4A1*	*CBP20*	*TBP1*	*GAPA*
		1.15 ± 0.32	1.16 ± 0.35	1.85 ± 0.56	1.87 ± 0.57	2.00 ± 0.52	2.25 ± 0.67	2.63 ± 0.79	2.72 ± 0.84	3.32 ± 0.77
Tissue	RSLF	*ACT7*	*TBP1*	*EF-1α*	*EIF4A1*	*CBP20*	*TUB1*	*eIF2*	*GAPB*	*GAPA*
		2.77 ± 0.79	2.87 ± 0.87	3.12 ± 0.92	4.03 ± 1.19	4.43 ± 1.34	4.81 ± 1.41	5.77 ± 1.64	9.32 ± 2.57	11.63 ± 2.95
	R	*GAPB*	*eIF2*	*TBP1*	*ACT7*	*GAPA*	*CBP20*	*EIF4A1*	*TUB1*	*EF-1α*
		0.70 ± 0.23	1.46 ± 0.46	1.59 ± 0.50	1.81 ± 0.53	1.86 ± 0.57	2.22 ± 0.70	2.64 ± 0.82	3.19 ± 0.98	3.33 ± 0.96
	S	*EF-1α*	*EIF4A1*	*eIF2*	*TBP1*	*CBP20*	*ACT7*	*GAPA*	*TUB1*	*GAPB*
		1.41 ± 0.42	1.52 ± 0.45	1.58 ± 0.44	1.70 ± 0.52	1.73 ± 0.53	1.83 ± 0.53	2.36 ± 0.57	2.80 ± 0.82	3.12 ± 0.84
	L	*TBP1*	*CBP20*	*GAPA*	*eIF2*	*EIF4A1*	*TUB1*	*GAPB*	*EF-1α*	*ACT7*
		1.34 ± 0.39	1.58 ± 0.48	1.79 ± 0.40	1.92 ± 0.52	2.01 ± 0.58	2.32 ± 0.69	2.35 ± 0.58	2.85 ± 0.84	3.32 ± 0.94
	F	*TBP1*	*CBP20*	*GAPB*	*ACT7*	*TUB1*	*EIF4A1*	*eIF2*	*GAPA*	*EF-1α*
		1.59 ± 0.47	1.86 ± 0.50	2.25 ± 0.59	2.57 ± 0.72	2.66 ± 0.70	2.85 ± 0.77	3.06 ± 0.83	3.14 ± 0.73	3.58 ± 1.05

* Nine candidate reference genes are ranked by the lowest values of the coefficient of variance (CV) and standard deviation (SD), and these values are shown as CV ± SD.

**Table 3 genes-13-01227-t003:** A comprehensive ranking of nine candidate reference genes integrated by RefFinder.

Group	Organ	Rank *								
		1	2	3	4	5	6	7	8	9
Total	RSL	*TBP1*(1.41)	*CBP20*(2.11)	*EIF4A1*(2.21)	*TUB1*(3.66)	*eIF2*(4.92)	*ACT7*(5.73)	*EF-1α*(6.19)	*GAPB*(8.00)	*GAPA*(9.00)
	R	*TBP1*(1.00)	*EIF4A1*(1.86)	*GAPB*(2.71)	*TUB1*(4.23)	*GAPA*(5.23)	*CBP20*(6.05)	*eIF2*(6.48)	*ACT7*(7.74)	*EF-1α*(9.00)
	S	*GAPB*(1.86)	*EIF4A1*(2.21)	*CBP20*(2.91)	*TBP1*(3.13)	*GAPA*(4.16)	*TUB1*(4.21)	*eIF2*(6.44)	*EF-1α*(8.00)	*ACT7*(9.00)
	L	*GAPB*(1.32)	*eIF2*(2.11)	*TBP1*(3.03)	*CBP20*(4.36)	*TUB1*(4.56)	*EIF4A1*(5.14)	*GAPA*(5.26)	*EF-1α*(7.48)	*ACT7*(9.00)
Cold	RSL	*TBP1*(1.86)	*eIF2*(1.93)	*EIF4A1*(2.91)	*TUB1*(3.66)	*CBP20*(3.72)	*ACT7*(5.23)	*EF-1α*(5.66)	*GAPB*(8.00)	*GAPA*(9.00)
	R	*TBP1*(1.73)	*TUB1*(1.86)	*eIF2*(2.34)	*CBP20*(3.34)	*GAPB*(4.43)	*EIF4A1*(5.42)	*GAPA*(7.48)	*ACT7*(7.48)	*EF-1α*(9.00)
	S	*EF-1α*(1.68)	*eIF2*(1.86)	*CBP20*(2.21)	*EIF4A1*(3.22)	*GAPA*(5.32)	*TBP1*(6.40)	*GAPB*(6.62)	*TUB1*(6.74)	*ACT7*(9.00)
	L	*TUB1*(2.11)	*eIF2*(2.45)	*GAPA*(2.94)	*GAPB*(3.31)	*TBP1*(3.46)	*EIF4A1*(4.74)	*CBP20*(5.63)	*ACT7*(7.48)	*EF-1α*(8.74)
Heat	RSL	*eIF2*(1.63)	*TBP1*(2.00)	*EIF4A1*(3.13)	*CBP20*(3.31)	*TUB1*(3.41)	*EF-1α*(5.48)	*ACT7*(6.74)	*GAPB*(8.00)	*GAPA*(9.00)
	R	*eIF2*(2.00)	*EIF4A1*(2.24)	*GAPA*(2.45)	*TBP1*(2.91)	*GAPB*(3.87)	*TUB1*(5.09)	*EF-1α*(7.24)	*CBP20*(8.13)	*ACT7*(8.24)
	S	*eIF2*(1.57)	*GAPA*(1.57)	*TBP1*(3.13)	*CBP20*(3.16)	*EIF4A1*(4.61)	*GAPB*(5.86)	*TUB1*(6.48)	*EF-1α*(8.00)	*ACT7*(9.00)
	L	*TBP1*(1.68)	*TUB1*(2.00)	*eIF2*(2.59)	*GAPA*(3.83)	*GAPB*(4.16)	*EIF4A1*(4.36)	*CBP20*(7.24)	*ACT7*(7.74)	*EF-1α*(9.00)
PEG	RSL	*EIF4A1*(1.41)	*TBP1*(1.68)	*CBP20*(3.57)	*TUB1*(4.23)	*ACT7*(4.40)	*EF-1α*(5.20)	*eIF2*(6.24)	*GAPB*(7.24)	*GAPA*(8.24)
	R	*GAPB*(1.32)	*EIF4A1*(2.06)	*eIF2*(2.11)	*TBP1*(4.00)	*CBP20*(4.61)	*ACT7*(6.19)	*GAPA*(6.48)	*EF-1α*(8.00)	*TUB1*(9.00)
	S	*EIF4A1*(2.24)	*eIF2*(3.13)	*GAPB* (3.83)	*ACT7*(4.28)	*CBP20*(4.45)	*TBP1*(4.68)	*GAPA*(4.90)	*TUB1*(5.01)	*EF-1α*(5.20)
	L	*TUB1* (1.86)	*EIF4A1*(2.21)	*GAPB*(2.45)	*TBP1*(3.13)	*eIF2*(3.34)	*GAPA*(5.73)	*CBP20*(7.00)	*ACT7*(8.00)	*EF-1α*(9.00)
CuSO_4_	RSL	*TBP1*(2.21)	*EF-1α*(2.34)	*ACT7*(2.63)	*EIF4A1*(2.91)	*CBP20*(4.43)	*TUB1*(4.73)	*GAPB*(5.45)	*eIF2*(7.48)	*GAPA*(9.00)
	R	*TBP1*(1.19)	*EF-1α*(1.41)	*GAPB*(3.22)	*eIF2*(4.12)	*EIF4A1*(5.44)	*TUB1*(5.73)	*ACT7*(6.09)	*GAPA*(8.00)	*CBP20*(9.00)
	S	*GAPB*(1.57)	*eIF2* (1.86)	*GAPA*(2.00)	*EIF4A1*(3.87)	*TBP1*(4.68)	*TUB1*(6.24)	*EF-1α*(6.65)	*CBP20*(7.74)	*ACT7*(9.00)
	L	*CBP20*(2.21)	*ACT7*(2.21)	*EF-1α*(2.45)	*GAPB*(2.91)	*TBP1*(4.16)	*eIF2*(4.68)	*TUB1*(6.44)	*GAPA*(7.74)	*EIF4A1*(9.00)
NaCl	RSL	*TBP1*(1.00)	*CBP20*(2.38)	*eIF2*(3.03)	*EIF4A1*(3.46)	*TUB1*(5.00)	*ACT7*(6.00)	*EF-1α*(6.05)	*GAPB*(7.48)	*GAPA*(9.00)
	R	*EIF4A1*(1.73)	*GAPA* (1.97)	*TUB1*(2.00)	*eIF2*(3.76)	*GAPB*(4.82)	*TBP1*(4.90)	*EF-1α*(7.24)	*CBP20*(7.74)	*ACT7*(9.00)
	S	*eIF2*(1.93)	*TBP1* (2.21)	*EF-1α*(2.91)	*GAPB*(3.72)	*GAPA*(4.28)	*ACT7*(5.24)	*CBP20*(5.62)	*EIF4A1*(7.09)	*TUB1*(7.35)
	L	*eIF2*(1.00)	*TBP1*(1.68)	*GAPB*(3.41)	*GAPA*(4.16)	*TUB1*(4.90)	*CBP20*(5.38)	*EIF4A1*(7.42)	*EF-1α*(7.74)	*ACT7*(8.45)
ABA	RSL	*EF-1α*(1.50)	*EIF4A1*(2.21)	*TBP1*(2.91)	*TUB1*(3.66)	*CBP20*(4.47)	*eIF2*(4.92)	*ACT7*(5.63)	*GAPB* (7.74)	*GAPA*(9.00)
	R	*EF-1α*(1.78)	*ACT7*(1.86)	*EIF4A1* (3.22)	*TBP1*(3.36)	*CBP20*(3.66)	*GAPB*(5.69)	*TUB1*(5.86)	*eIF2*(7.74)	*GAPA*(9.00)
	S	*GAPB*(1.73)	*eIF2*(1.73)	*TBP1*(2.66)	*EF-1α*(3.13)	*TUB1*(4.95)	*EIF4A1*(5.63)	*CBP20*(7.24)	*ACT7*(7.35)	*GAPA*(8.24)
	L	*GAPB*(1.57)	*EF-1α*(2.21)	*eIF2*(3.08)	*TUB1*(3.34)	*EIF4A1*(3.46)	*TBP1*(5.09)	*ACT7*(6.74)	*CBP20*(8.00)	*GAPA*(9.00)
MeJA	RSL	*ACT7*(1.97)	*EIF4A1*(2.00)	*TBP1*(3.08)	*TUB1*(3.74)	*eIF2*(3.96)	*EF-1α*(4.12)	*CBP20*(5.73)	*GAPB*(8.00)	*GAPA*(9.00)
	R	*TUB1*(1.41)	*GAPB*(2.45)	*ACT7*(3.36)	*EF-1α*(3.46)	*eIF2*(4.05)	*TBP1*(5.09)	*EIF4A1*(5.44)	*GAPA*(7.48)	*CBP20*(9.00)
	S	*eIF2* (1.57)	*GAPB* (1.86)	*TUB1*(2.06)	*GAPA*(4.43)	*EIF4A1*(5.92)	*ACT7*(6.18)	*CBP20*(6.45)	*TBP1*(6.85)	*EF-1α*(7.11)
	L	*TUB1*(1.00)	*EIF4A1*(1.86)	*GAPB*(3.13)	*eIF2*(3.87)	*TBP1*(4.47)	*ACT7*(6.70)	*CBP20*(6.70)	*GAPA*(7.97)	*EF-1α*(8.45)
SA	RSL	*EIF4A1*(1.50)	*CBP20*(2.21)	*TBP1*(3.36)	*ACT7*(3.66)	*TUB1*(3.87)	*eIF2*(4.58)	*EF-1α*(5.86)	*GAPB*(8.00)	*GAPA*(9.00)
	R	*GAPB*(1.41)	*eIF2*(1.68)	*EIF4A1*(3.56)	*TBP1*(3.87)	*TUB1*(4.05)	*ACT7*(5.66)	*CBP20*(6.45)	*GAPA*(7.45)	*EF-1α*(8.45)
	S	*TUB1*(2.11)	*EIF4A1*(2.51)	*CBP20*(2.71)	*GAPA*(3.64)	*GAPB*(4.43)	*TBP1*(5.20)	*ACT7*(5.86)	*EF-1α*(6.26)	*eIF2*(6.90)
	L	*eIF2*(1.32)	*ACT7*(1.41)	*GAPB*(2.71)	*GAPA*(4.60)	*CBP20*(5.62)	*TBP1*(6.64)	*EIF4A1*(6.74)	*EF-1α*(7.11)	*TUB1*(7.35)
Tissue	RSLF	*EIF4A1*(1.41)	*TBP1*(1.86)	*ACT7*(3.16)	*TUB1* (4.36)	*eIF2*(4.41)	*EF-1α*(5.12)	*CBP20*(5.18)	*GAPB*(8.00)	*GAPA*(9.00)
	R	*TBP1*(1.97)	*GAPB*(2.00)	*EIF4A1*(2.82)	*TUB1*(3.76)	*ACT7*(3.94)	*GAPA*(5.63)	*eIF2*(6.18)	*EF-1α*(7.17)	*CBP20*(7.44)
	S	*EIF4A1*(2.00)	*TBP1*(2.34)	*eIF2*(2.45)	*GAPA*(3.25)	*EF-1α*(4.60)	*CBP20*(5.23)	*ACT7*(5.86)	*GAPB*(6.64)	*TUB1*(8.74)
	L	*EIF4A1*(2.24)	*GAPA*(2.74)	*TBP1*(2.78)	*eIF2*(2.83)	*TUB1*(4.74)	*CBP20*(4.74)	*EF-1α*(5.57)	*GAPB*(5.63)	*ACT7*(9.00)
	F	*TBP1*(1.41)	*GAPB*(1.73)	*TUB1*(2.99)	*EIF4A1*(3.98)	*CBP20*(4.36)	*ACT7*(4.95)	*eIF2*(7.24)	*GAPA*(7.44)	*EF-1α*(9.00)

* RefFinder exploits computational programs (such as BestKeeper, geNorm, NormFinder, or the comparative delta-Ct method) to rank and compare candidate reference genes. The values following the genes indicate the geometric mean of the attributed weights measured by this software for the overall final ranking.

## Data Availability

RNA-seq datasets used in this study can be found in online repositories. The names of the repository/repositories and accession number(s) can be found below: https://www.ncbi.nlm.nih.gov/, PRJNA848854.

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
