# Peer review of "Identification of Reference Genes for RT-qPCR Analysis in Gleditsia microphylla under Abiotic Stress and Hormone Treatment"

_genes, 2022, doi:10.3390/genes13071227_

Round 1

Reviewer 1 Report

no suggestion.

Author Response

Dear Reviewer,

    We thank you for your kind approval for our manuscript (genes-1795704). This paper has undergone English language editing by MDPI, and a certificate (46724) has been supplied. As for the detailed changes, please see the attachment.

Reviewer 2 Report

Dear Sir,

The scientific writing of the manuscript is satisfactory. The manuscript can be published with minor corrections in the introduction part only.

regards 

Author Response

Dear Reviewer,

    We thank you for your kind comments and suggestions for our manuscript (genes-1795704). This paper has undergone English language editing by MDPI, and a certificate (46724) has been supplied.

List of Responses:

Point 1: The scientific writing of the manuscript is satisfactory. The manuscript can be published with minor corrections in the introduction part only.

Response 1: In the introduction part, more details were added to the aims of this work (Page 2), which make it easier for understanding. As for the detailed changes, please see the attachment.

Reviewer 3 Report

Gleditsia microphylla is an important galactomannan gums source plant with characteristics of drought resistance, barren tolerance, and good adaptability. However, underlying molecular mechanisms of the biological process have not been fully understood yet. Real-time quantitative PCR (RT-qPCR) is an accurate and convenient method to quantify the gene expression level and transcription abundance with suitable reference genes. This study aimed to screen the best internal reference genes in Gleditsia microphylla under abiotic stresses, hormone treatments, and different tissues. Based on the transcriptome data, twelve candidate reference genes were selected, and ultimately, nine of them were further evaluated by geNorm, NormFinder, BestKeeper, and RefFinder algorithms. These results identify that TBP1 and EIF4A1 were the two most stable reference genes, and GAPA and GAPB were the two most unstable reference genes across all samples under given experimental conditions. Meanwhile, the most stable reference genes varied flunctantly among different groups and tissues. Therefore, this study suggests that it is better to use a specific reference gene for a particular case rather than using a common reference gene.

Based on these findings, I suggest some shortcomings to improve this investigation.

1-     Introduction: More details should be added to the aim of the work.

2-     Results: Please add the statistics on the figures related to boxplots, such as significant differences,,,,,,etc.

3-     Discussion: Please discuss the genes under different environmental stimuli that are related to Gleditsia microphylla.

4-     Conclusion: More information about the findings should be added.

5-     Several grammatical errors are involved in this manuscript.

Author Response

Dear Reviewer,

    We thank you for your kind comments and suggestions for our manuscript (genes-1795704). This paper has undergone English language editing by MDPI, and a certificate (46724) has been supplied. As for the detailed changes, please see the attachment.

List of Responses:

Point 1: 1-Introduction: More details should be added to the aim of the work.

Response 1: In the introduction part, more details were added to the aims of this work (Page 2), which make it easier for understanding.

Point 2: 2-Results: Please add the statistics on the figures related to boxplots, such as significant differences,,,,,,etc.

Response 2: In the results part (3.2 Expression Profiling of Candidate Reference Genes), intra- and intergroup statistical analyses of Ct values were added (Page 5), which will present an overall variation trend for all candidates.

Point 3: 3-Discussion: Please discuss the genes under different environmental stimuli that are related to Gleditsia microphylla.

Response 3: ACO1 and CSD2 cloned in Gleditsia microphylla by our team have high sequence identity with homologs in other plant species, which means they would have similar biological functions in response to environmental stimuli. It is the reason we choose them as targets for evaluation. And a brief introduction of these genes can be found in Materials and Methods 2.6. Related discussion was added (Page 21).

Point 4: 4-Conclusion: More information about the findings should be added.

Response 4: In conclusion part, based on the topic identification of internal reference genes for RT-qPCR in Gleditsia sinensis, we stated the most stable and unstable reference genes under experimental conditions, and suggested the most stable reference genes to choose in application. If the readers want to carry out RT-qPCR analysis in Gleditsia sinensis, they can easily find the appropriate reference genes. We think there is no more valuable information can be added.

Point 5: 5- Several grammatical errors are involved in this manuscript.

Response 5: This paper (genes-1795704) has undergone English language editing by MDPI, and a certificate (46724) has been supplied. As for the detailed changes, please see the attachment.
